# Development and Characterization of a Novel Sustainable Probiotic Goat Whey Cheese Containing Second Cheese Whey Powder and Stabilized with Thyme Essential Oil and Sodium Citrate

**DOI:** 10.3390/foods11172698

**Published:** 2022-09-04

**Authors:** Manuel Mântua Esteves Garcia, Carlos José Dias Pereira, Ana Cristina Freitas, Ana Maria Pereira Gomes, Maria Manuela Estevez Pintado

**Affiliations:** 1Centro de Biotecnologia e Química Fina (CBQF)-Laboratório Associado, Escola Superior de Biotecnologia, Universidade Católica Portuguesa, Rua Diogo Botelho 1327, 4169-005 Porto, Portugal; 2Escola Superior Agrária-Politécnico de Coimbra (ESAC-IPC), Bencanta, 3040-316 Coimbra, Portugal

**Keywords:** probiotic, second cheese whey, functional food, essential oil, Requeijão, organoleptic profile

## Abstract

Probiotic goat whey cheeses with added second cheese whey powder (SCWP) were developed, resulting in creamy and spreadable products. The products contained *Lactobacillus rhamnosus* and *Bifidobacterium animalis*, as well as thyme essential oil and sodium citrate. Matrices of probiotic whey cheeses, with and without additives, were produced and stored at 5 °C for 21 days. Microbial and chemical profiles were evaluated weekly. The composition of the optimum matrix, formulated with whey cheese, probiotic culture, SCWP, thyme essential oil and sodium citrate (WCPSTC) was, expressed in % (*w*/*w*): protein (10.78 ± 0.08), fat (7.59 ± 0.03), dry matter (25.64 ± 0.13), ash (2.81 ± 0.02) and lactose (3.16 ± 0.04). Viable cell numbers of both probiotic cultures in matrix WCPSTC remained above 10^7^ CFU g^−1^. This finding is of the utmost importance since it proves that both probiotic bacteria, citrate and thyme essential oil can be combined in order to increase the shelf-life and functional value of dairy products. All matrices’ pH values decreased during storage, yet only matrix WCPSTC remained above 5.0 pH units. The results indicated that the development of a probiotic whey cheese incorporating a dairy by-product, SCWP, is possible without compromising its chemical, microbiological or sensorial stability.

## 1. Introduction

The current food market is being driven by highly demanding consumers who are becoming not only more environmentally conscious but also very aware of the impact of diet on their health [1]. In response, the food industry must be able to produce functional foods of high nutritional value while simultaneously assuring natural resources sustainability. The dairy industry is one of the most polluting industries, generating about 0.2–10 L of effluents per L of processed milk [2]. These effluents are generally comprised of wastewater and by-products such as cheese whey and second cheese whey, which carry a biological oxygen demand between 0.6 and 60 g L^−1^ [3].

Large amounts of cheese whey are generated worldwide as a by-product of cheese production. Cheese whey contains more than half of the solids present in the original whole milk, including whey proteins, lactose, water-soluble vitamins and minerals [4]. Due to its nutritional value, around 50% of cheese whey generated worldwide is treated and re-incorporated into various food products [5]. In some Mediterranean countries, a fraction of this whey is used to produce whey cheeses, locally known as “Requeijão” (Portugal), “Requéson” (Spain) or “Ricotta” (Italy) [6]. Whey cheese is a traditional product manufactured via thermal processing of whey that allows retrieval of most of the original dry matter of cheese whey [7]. However, its traditional production method is a very low-yield process (about 2–3%), which, in turn, generates large quantities of a by-product known as second cheese whey (SCW) [8].

Second cheese whey is the final by-product of two consecutive cheese-making processes and is commonly regarded as a real waste, yet it still retains about 60% of the dry matter content of the original whey [8,9]. Regardless of its nutritional value and abundance of supply, SCW has not yet been sufficiently studied and explored, which may explain its limited industrial application so far. Research has mostly focused on effluent treatment technologies for SCW disposal [8,10], lactose recovery [11] and SCW conversion in bio-fuel [12]. Regarding SCW’s use in food products, a relatively low number of studies have been published on its potential, either as a source of bioactive peptides [13] or as a suitable matrix for probiotic beverages [14,15].

Regarding probiotics, these are defined as “live microorganisms which when administered in adequate amounts confer a health benefit on the host” [16]. Although the concentration of probiotic microorganisms that provides health benefits is usually strain- and host-dependent, levels equal to or greater than 10^7^ CFU g^−1^ have been suggested as the minimum concentration for a positive effect of probiotic bacteria [17].

With the aim of adding value to whey cheese products, probiotic whey cheeses were originally proposed by Madureira et al. [18]. Several probiotic strains of *Lactobacillus* and *Bifidobacterium* were incorporated in whey cheeses where, in most cases, high survival rates above 10^8^ CFU g^−1^ were observed [19]. As such, whey cheese presents itself as a viable dairy matrix for probiotic strain delivery. However, several bottlenecks of organoleptic nature have been pointed out regarding this type of dairy product. Whey cheese, besides its bland taste and grainy texture, often labeled as an old-fashioned “diet food” [20], becomes excessively acidic during storage when probiotic bacteria are added as a result of their organic acids production [21]. In order to counteract these organoleptic barriers, sodium citrate, an acidity regulator, has been used in fermented dairy products [22]. Plant essential oils, which have widespread use as a natural food additive [23], could also be added to mask off-flavors and increase the sensorial complexity of probiotic dairy products. Furthermore, both these additives exhibit antimicrobial activity against several contaminants and pathogen bacteria [24,25,26].

In view of the above considerations, this research effort focused on the development of a spreadable whey cheese-based product that could: (i) assure probiotic bacteria survival without microbial contamination; (ii) improve sensory quality and consistency throughout storage; (iii) incorporate in its matrix a dairy by-product (SCWP) and thyme essential oil without adverse effects and (iv) assure metabolic activity of probiotic bacteria but with stabilized pH throughout the refrigerated storage period.

By considering the beneficial nutritional profile of whey cheese and its potential as a probiotic carrier, a new strategy was thought and applied. The combination of traditional whey cheese “Requeijão” with SCWP, probiotic cultures and ingredients improving the organoleptic characteristics of the final product was studied. In addition to the development of a new functional food where nutrition and biological activity go hand in hand, this approach enables minimizing the environmental impact of the cheese whey and specially SCW by-products, assuring a sustainable cheese supply chain. Furthermore, a new added-value role was identified for a secondary by-product helping to keep the value of products and minimizing waste.

## 2. Materials and Methods

### 2.1. Microorganism Sources

*Bifidobacterium animalis* subsp. *lactis* BB-12^®^ (BB-12) and *Lactobacillus rhamnosus* Rosell^®^-11 (R-11) were obtained as lyophilized cultures from Christian Hansen (Hoersholm, Denmark) and Lallemand (Montreal, QC, Canada), respectively. Prior to inoculation of whey cheese matrix, cultures of BB-12 and R-11 were grown first in modified Man Rogosa and Sharpe (MRS) (Biokar, Beauvais, France) broth supplemented with filter-sterilized 0.5 g L^−1^ of L-cysteine-HCl (Lab M, Lancashire, United Kingdom) at 37 °C for 24 h in an anaerobic chamber (Don Whitley anaerobic chamber). Thereafter, both strains were cultured again in 10% (*w*/*v*) reconstituted skim milk (Sigma-Aldrich, St Louis, MO, USA) for another 24 h at 37 °C in an anaerobic chamber.

Contaminant bacteria *Bacillus cereus* NCTC 2599, *Listeria monocytogenes* Scott A-ATCC 15313, *Staphylococcus aureus* ATCC 6538, *Escherichia coli* ATCC 8739 and *Salmonella enterica* serovar *enteritidis* ATCC 13,076 used for antimicrobial assays were obtained from a collection of CBQF-Universidade Católica Portuguesa (Porto, Portugal). Before their use to determine minimum inhibitory concentrations (MIC), bacteria were grown in Mueller–Hinton broth (Biokar, Beauvais, France) at 37 °C for 24 h.

### 2.2. Second Cheese Whey Processing

Goat SCW was supplied by Tété II, Produtos Lácteos, Lda., (Lousa, Portugal). After the reception, the SCW was filtered using a cloth filter in order to separate suspended particulate material and then processed at 45–50 °C using a filtration pilot plant (Proquiga, Spain) equipped with an organic Parker™ membrane, model SD3838 BS 03S, 6.3 m^2^ filtering area, and 10 kDa cutoff. The transmembrane pressure was held at 3.5–4.5 bar. After the first concentration step, a volumetric concentration factor (VCF = volume feed/volume retentate) of 20,225 L of water was added to the 25 L of retentate, and a second concentration step was performed in order to obtain a final volume of 25 L diafiltrated retentate. Finally, this retentate was submitted to reverse osmosis using a pilot plant equipment (ORM, Lisboa, Portugal) equipped with a membrane Seawater model 2.5 S (Advanced Structures Inc., Escondido, CA, USA) using a VCF of 5. The final concentrated retentate was sterilized at 120 °C for 15 min and then freeze-dried in a Labconco Liph-Lock 12 stoppering tray dryer (Labconco, Kansas, MO, USA). The obtained powder had the following composition in % (*w*/*w*): total solids (96.50), protein (63.15), fat (25.80), other solids (5.09) and ashes (2.45).

### 2.3. Manufacture of Probiotic Whey Cheese

Goat whey cheese “Requeijão” obtained from whey derived from goat fresh cheese manufacture was provided by Tété II company and kept at 5 °C in aseptic conditions for a maximum of 48 h until probiotic whey cheese production. Goat whey cheese and SCWP at 5% (*w*/*w*) were homogenized for 10 min, followed by pasteurization for 30 min at 100 °C using a Bimby^®^ TM6 (Vorwerk Thermonix, Wuppertal, Germany). The resulting mixture was then cooled to 30 °C under continuous mixing in order to produce a creamy and spreadable texture and subsequently inoculated with the probiotic co-culture at 10% (*v*/*w*), in equal volume proportions, i.e., 5% (*v*/*w*) of both BB-12 and R-11 cultures previously grown as stated above. The initial concentration of bacterial cultures, 9.30 log CFU g^−1^ and 9.19 log CFU g^−1^ for BB-12 and R-11, respectively, allowed an initial probiotic viable cell concentration above 10^7^ CFU g^−1^ in the whey cheese mixture. From this point, the whole batch was divided into three portions: Portion I without further additives—matrix WCPS; Portion II with the addition of thyme essential oil (Socidestilda, Lisbon, Portugal), 0.025% (*w*/*w*)—matrix WCPST; Portion III with the addition of thyme essential oil (EO), 0.025% (*w*/*w*) and sodium citrate (Merck, Darmstadt, Germany), 0.8% (*w*/*w*)—matrix WCPSTC. Prior to the addition of the whey cheese matrix, sodium citrate was dissolved in sterile water and filter-sterilized (0.22 µm). A batch of probiotic whey cheese without incorporation of SCWP nor other additives was produced following the same processing and inoculation conditions—matrix WCP. In addition, goat whey cheese, without any processing or inoculation, matrix WC was used as a control “Requeijão” in order to allow comparative analysis of the effect of additives and probiotic culture in whey cheese matrices. All matrices were equally distributed into 100 mL sterile sample containers, which were immediately sealed (in order to simulate closed packages) and stored at 5 °C for up to 21 days.

### 2.4. Microbiological Analysis

The viable cell numbers of both probiotic bacteria and eventual contaminants were assessed by preparing serial decimal dilutions with 0.1% (*w*/*v*) peptone (Becton Dickinson, Franklin Lakes, NJ, USA) and 0.85% (*w*/*v*) NaCl (Fluka, Buchs, Switzerland) water. Whey cheese matrices were sampled in duplicate (5-g aliquot). For quantification and detection of probiotic bacteria and contaminants, dilutions up to −8 were plated, in duplicate, on the following media: Plate Count Agar (VWR, Leuven, Belgium) for contaminant aerobic mesophilic bacteria; Man Rogosa and Sharpe Agar (MRS) (Biokar, Beauvais, France) for viable cells of *L. rhamnosus* R-11; broth selective medium (BSM) (Sigma-Aldrich, St Louis, MO, USA) supplemented with bacteriologic agar (Biokar, Beauvais, France) at 15 g L^−1^ for *B. animalis* BB-12; Rose Bengal Chloramphenicol Agar Base (RBCA) (Lab M, Lancashire, United Kingdom) for detection of yeasts and molds; Violet Red Bile Glucose Agar (VRBGA) (VWR, Leuven, Belgium) for detection of coliform bacteria and *Enterobacteriaceae*; and Bacillus Cereus Agar (base acc. to Mossel) supplemented with egg-yolk emulsion (50 mL L^−1^) and Polymyxin B at 100 IU mL^−1^ reconstituted in 5 mL of sterile water (Biokar, Beauvais, France) for detection of *Bacillus cereus*. All the aforementioned media were plated using the Miles and Misra method [27], with the exception of VRBGA (pour plate technique) and RBCA (spread plate technique). The results were presented as a log of mean colony-forming units (CFU) per gram of whey cheese matrix. Incubation conditions of the inoculated plates were as follows: PCA, 30 °C for 48 h; MRS, 37 °C for 48–72 h; BSM, 37 °C for 48–72 h in anaerobiosis using BBL-Gaspak anaerobic system (Becton Dickinson, Franklin Lakes, NJ, USA); Bacillus Cereus Agar, 37 °C for 24 h; VRBGA, 30 °C for 24 h followed by another 24 h at 37 °C; and RBCA, for 5 days at 30 °C.

Antimicrobial assays were performed according to Wiegand et al. [28], following the broth dilution method and using a 96-well microtiter plate. Mueller–Hinton broth (MHB) was used as the non-selective growth medium for contaminant and probiotic bacteria. Decreasing concentrations of essential oil and sodium citrate, with a maximum concentration of 1.0 mg mL^−1^ and 32 mg mL^−1^, respectively, were dissolved in MHB and added to the microtiter plate. Inoculation of bacteria in the wells was performed in order to achieve an inoculum concentration between 10^5^ and 10^6^ CFU mL^−1^. Bacterial growth was assessed after incubation for 16–24 h at 37 °C either in anaerobic conditions (for assays containing *B. animalis* BB-12) or aerobic conditions for the remaining bacteria. The minimum Inhibitory Concentration (MIC) value was considered to be the lowest concentration of each compound that prevented the visible growth of each bacterium.

### 2.5. Physicochemical Analysis

Moisture content and ash content of whey cheese matrices were determined according to AOAC International methods (1995) [29]. Determination of pH values of whey cheese samples was made with an electrode penetration probe connected to a MicropH 2002 apparatus (Crison, Barcelona, Spain). The titratable acidity of samples was determined by titration with 0.1 M NaOH solution using phenolphthalein at 0.1% (*w*/*v*) (Fluka, Buchs, Switzerland) as an acid-base indicator. Acidity was expressed as a percentage of lactic acid. Total nitrogen content was determined using the Kjeldahl method. The method was followed using a Digestor 2012 (FOSS analytical, Hilleroed, Denmark) and a Kjeldahl Distillation Unit Velp UDK 129 (Velp Scientifica, Usmate, Italy). Total protein content was estimated by multiplying the total nitrogen content of the samples by a factor of 6.38.

For the determination of total fat in samples, an extraction process was adapted from Hara and Radin methodology [30]. Thereby, 18 mL of hexane/isopropanol was added (3:2) to 1 g of sample; the mixture was shaken for 3 min in a vortex under maximum velocity. The solution was then filtered through filter paper Whatman^®^ No. 42, and 2 mL of the hexane/isopropanol solution was used to wash the filter paper. Then, 12 mL of sodium sulfate at 66.6 g L^−1^ (Merck, Darmstadt, Germany) was added, shaken for 2 min in a vortex and then maintained at rest until the phase separation was achieved. The lipid phase of the sample was collected in 50 mL flasks and evaporated through a rotavapor evaporator (BÜCHI, Flawil, Switzerland) at 40 °C. The difference in weight between the empty flask and the flask after evaporation of the lipid phase was considered to be the fat content of the sample. All reagents used were of analytical grade, and all analyses were carried out in triplicate.

### 2.6. Determination of Sugars and Organic Acids

Replicated samples of whey cheese, taken at 0, 7, 14 and 21 days of storage, were assayed for lactose and lactic acid content. Quantification was performed by HPLC based on calibration curves prepared in advance with appropriate chromatographic standards for lactose and lactic acid (Sigma-Aldrich, St Louis, MO, USA).

For the preparation of samples, an aliquot of 2 g was homogenized with 10 mL of 13mM H_2_SO_4_ (Sigma-Aldrich, St Louis, MO, USA) in an Ultra-Turrax^®^ IKA T25 Digital (Staufen, Germany) for 3 min at 18,000 rpm. Subsequently, the samples were centrifuged at 4000 rpm for 10 min at 4 °C and then filtered through Whatman^®^ No. 42 filter paper. Immediately prior to HPLC injection, samples were again filtered through a 0.22 µm syringe filter (Sartorius, Göttingen, Germany). Samples thus treated were injected in an ion exchange aminex column HPX-87H (Bio-Rad, Hercules, CA, USA) maintained at 40 °C with refractive index (RI) and ultra-violet (UV) detector (220 nm) for sugar and organic acid detection, respectively. The mobile phase used was 13 mM H_2_SO_4_, regenerating the column with 25 mM H_2_SO_4_ when necessary. The flow rate was defined at 0.6 mL min^−1^. The run time of duplicate samples was 40 min, and the injection volume was 50 µL.

### 2.7. Textural Analysis

Texture properties of the several matrices at 0 and 21 days of storage were assessed via measurement of the force-time curve with a TA.XT apparatus (Stable Micro Systems, Surrey, UK). The probe used was P/5 (a 5 mm diameter cylindrical device) attached to a 5 kg load cell, and tests were performed directly in the containers (in triplicate). Calibration of the cell was performed with a 2 kg weight. A typical “mastication test” testing profile was followed, which involves two consecutive compressions at controlled room temperature (25 ± 1 °C). The compression distance used was 10 mm with a trigger force of 1.0 g at a test speed of 5 mm s^−1^. This test made it possible to measure several attributes, such as hardness, gumminess, cohesiveness and adhesiveness.

### 2.8. Sensory Analysis

Before the preparation of the final products, sensory analysis was conducted in order to determine the optimum aromatized whey cheese formulation. Therefore, prior to the manufacture of probiotic matrices, four types of aromatized whey cheese batches with different additives were tasted by non-trained participants. The base formulation of the aromatized whey cheeses was composed of whey cheese, 85% (*w*/*w*); SCWP, 5% (*w*/*w*); and probiotic culture, 10% (*v*/*w*). The formulations included in the sensory analysis were aromatized with thyme, garlic and sage essential oil, added at 0.025% (*w*/*w*) concentration, resulting in the following four distinct batches of aromatized probiotic whey cheese: A (thyme); B (thyme + garlic + sage); C (thyme + sage); D (thyme + garlic). The panel totalized 40 people, 25 female and 15 male, with an average age of 32 years. After confirmation of microbiological safety, samples of the four batches were given to taste on top of unflavored crackers, with water being provided to clean the palate between tastings. Attributes such as texture, flavor and overall acceptability were evaluated using a 9-point hedonic scale (in which 1 corresponds to “very bad” and 9 to “very good”). On the other hand, aroma, saltiness and acidity were evaluated on a qualitative scale of intensity from barely perceptible (weak) to overpowering (strong).

### 2.9. Statistical Analysis

One-way analysis of variance (ANOVA) was carried out with SPSS v. 26.0. (Chicago, IL, USA) to determine the effect of probiotic culture, SCWP and organoleptic additives in whey cheese physicochemical and microbiological profiles. Tukey’s HSD test, at the 5% level of significance, was applied to the experimental results concerning each type of matrix. Correlations between physicochemical parameters, lactose and lactic acid contents, and texture attributes were checked via Pearson’s test to a 1% significance level.

## 3. Results and Discussion

The major goal of the improvement process for probiotic whey cheese was to tackle issues of organoleptic, environmental and nutritional nature. Organoleptically, the addition of essential oil and sodium citrate could overcome the excess of acidity while adding flavor to the probiotic whey cheese. Additionally, through the incorporation of a dairy by-product, SCWP, the outcome should provide an environmentally sustainable functional food product of high nutritional value.

### 3.1. Selection of Essential Oils and Sodium Citrate Content

In order to address the sensory issues detected in probiotic whey cheese [21], essential oils were screened for added flavor and aroma (results not shown), with three being pre-selected for sensory analysis: thyme, sage and garlic essential oils. Despite being focused on sensory properties, their addition also had the intention of increasing the microbial safety of probiotic whey cheese, as some essential oils exhibit antimicrobial activity [23]. In order to tackle the acidity issue, sodium citrate was tested as a possible solution for controlling the acidification process, which, if successful, would significantly increase the organoleptic consistency of the product throughout storage.

Before conducting sensory analysis of possible whey cheese formulations, MIC determination was instrumental in defining both the concentration of EO to be used and which formulations would be pertinent to study further. MIC results present in Table 1 clearly distinguished the antibacterial activity of thyme EO, to the detriment of garlic and sage EO, with values ranging from 0.125 mg mL^−1^ for *S. aureus* and 0.5 mg mL^−1^ for *B. cereus* and *S. enteritidis*, results that are in agreement with those found by other authors [24,31,32]. Consequently, all whey cheese matrices produced for sensory analysis had thyme EO in their base formulation. Furthermore, a concentration of 0.25 mg mL^−1^ was found to be ideal, as it simultaneously prevents the growth of important contaminant bacteria and allows the growth of probiotic bacteria *B. animalis* BB-12 and *L. rhamnosus* R-11. Moritz and coauthors [33] presented similar results while screening several EOs for a probiotic yogurt containing *L. rhamnosus.* In that study, a concentration of 0.04% (*v*/*v*) scored highest in sensorial acceptance, while MIC for *L. rhamnosus* was between 0.2 and 0.4 % (*v*/*v*).

Sodium citrate MIC results showed minimal interference of this additive in probiotic bacteria growth, only inhibiting *B. animalis* BB-12 and *L. rhamnosus* R-11 at 16.0 mg mL^−1^ and 32.0 mg mL^−1^, respectively. Furthermore, when combined with thyme EO, sodium citrate synergistically reduces the MIC value for *S. enteritidis*.

In parallel with the antimicrobial effect of essential oils, the formulations with thyme oil, alone and combined with the other two essential oils (garlic and sage), were screened by a consumer panel. The results of the organoleptic assessment of the various flavored whey cheese matrices are depicted in Table 2 and Figure 1. Sensory analysis results express a positive tendency towards formulation A, with overall acceptance and flavor scoring significantly higher than other whey cheeses, with most of the panel claiming that this formulation had a pleasant and adequate aroma. Formulation B was mainly criticized for being too intense in terms of aroma and flavor, scoring the lowest of the four in most attributes. On the other hand, saltiness, acidity and texture were perceived by the panel as indistinguishable between the whey cheeses, yet in accordance with their expectation. In agreement with these, a study conducted by El-kholy et al. [34] also showed good sensorial acceptance of soft cheese with added thyme essential oil despite the higher concentration used, 0.1% (*w*/*w*).

When considering the results of both sensory analysis and MIC determination, formulation with thyme EO at a concentration of 0.25 mg mL^−1^ and sodium citrate at 8.0 mg mL^−1^ was considered the most favorable formulation for flavoring and enhancing microbial safety of probiotic whey cheese.

### 3.2. Viability of Probiotics in Whey Cheeses and Acidity throughout Storage Time

The evolution of viable cell numbers of probiotic strains *B. animalis* BB-12 and *L. rhamnosus* R-11 in the four whey cheese matrices, namely, WCP, WCPS, WCPST and WCPSTC, are depicted in Table 3. In all matrices, total viable cell numbers of probiotic strains increased up to ca. 0.5 log cycles during the 21-day storage period, yet values for matrices WCP and WCPS were significantly higher than WCPST and WCPSTC in almost one log cycle (*p* < 0.05). Only *B. animalis* BB12 exhibited limited growth in the presence of thyme EO and sodium citrate; however, these additives did not compromise its survival during 21 d storage.

Regarding detection of contaminants, aseptic conditions applied resulted in no external contamination, i.e., not found in the sample dilution 10^−1^, in all matrices of whey cheeses throughout storage, with the exception of mesophilic bacteria counts in the PCA growth medium. Nevertheless, the results of PCA cell counts were compared to the number of CFUs in MRS agar, and no statistically significant difference was found between growth media (data not shown), from which we can infer that both datasets belong to viable cells of *L. rhamnosus* R-11.

Results for viable cell numbers listed in Table 3 presented an average of 8.24 log CFU g^−1^ on the final day of storage, a value slightly below the one found by Madureira et al. [21]: an average of 9.27 log CFU g^−1^ at 21 days of storage. Nevertheless, it must be considered that a co-culture was used instead of a single probiotic strain, as was the case of the mentioned authors. However, the microbial concentration of probiotic bacteria remained above 10^7^ CFU g^−1^, which has been stated as the minimum required for a functional probiotic product [16,35]. These probiotic strains reveal technological adequacy as they exhibit an intrinsic ability to maintain high viable cell populations but also a capacity to withstand additives that are essential for desirable organoleptic features and for potentiating microbial safety.

The acidification process is expressed in terms of pH value and percentage of lactic acid in Table 3. Non-inoculated whey cheese (matrix WCP) presented pH 6.12 ± 0.01 and titratable acidity (%) 0.20 ± 0.00.

A drop in pH values was uniform in all whey cheese matrices, decreasing around 1.0 unit, regardless of initial probiotic cell concentration or formulation. Similar findings were found in probiotic “Minas Frescal” cheese (semi-soft) supplemented with *L. acidophilus* [36]. However, absolute pH values exhibited significant differences between matrices, with WCPSTC being the most distinct from the rest. While WCP, WCPS and WCPST exhibited a final pH value of around 4.8 due to uncontrolled acidification, it is apparent the positive effect that sodium citrate had on whey cheese matrix, with a pH value ca. 5.1. As reported previously, a negative sensory characteristic of probiotic whey cheese is the notorious acidity that becomes evident from 14 days storage onward, when pH drops significantly below 5.0 [21]. The addition of sodium citrate was revealed to be effective in controlling acidification during 21 days of storage.

Titratable acidity values expressed a similar expected tendency, increasing from an initial average value of 0.45% to a final value of 1.2% at 21 days of storage, with a concomitant drop in pH value. Emphasizing this relation, values of acidity and pH revealed very good linear correlation at (*p* < 0.01): WCP (r = −0.966); WCPS (r = −0.956); WCPST (r = −0.991); WCPSTC (r = −0.989). Little differences can be observed between formulations, except for WCPSTC, due to its formulation, which exhibits a significantly lower value than WCPST. No significant correlation was observed between viable cell concentration and titratable acidity or pH.

### 3.3. Physicochemical Profile of Whey Cheeses

The physiochemical parameters pertaining to non-inoculated whey cheese and the four inoculated matrices manufactured with several additives are represented in Table 4. Overall, whey cheeses compositions are in agreement with other types of whey cheeses, e.g., dried whey cheese produced with “Feta” cheese whey (Greece) [37], “Bracka skuta” whey cheese (Croatia) [38] and “Ricotta” (Italy) [39]. However, some differences were detected, mainly in the lower fat content of our whey cheese in comparison to others, which can be attributed to the different whey origins (e.g., bovine, ovine, caprine) and manufacturing processes [4].

Regarding the comparative analysis of whey cheese composition, the results obtained were very similar among the different formulations, which is understandable given the almost identical base formulation of the matrices. Nevertheless, some variations were observed in dry matter content, ash and total protein despite its nominal value being very similar between formulations, which can be explained by the addition of dissolved additives (sodium citrate, probiotic culture) and can contribute to a matrix dilution factor. As expected, the increase in fat content in WCPS compared to WC and WCP is associated with the addition of SCWP. Despite the observed increase in WCPS, its fat content was not significantly different from WC and was significantly lower than that of WCPST and WCPSTC. It should be mentioned that the WC sample was not inoculated with the starter culture. On the other hand, the combined addition of SCWP and essential oil is responsible for the higher fat content of WCPST and WCPSTC formulations. However, the increase in fat content of WCPST and WCPSTC relative to WCPS is just associated with the addition of the thyme essential oil, although the total amount of essential oils added was inferior to the final value observed in Table 4. The presence of EO in WCPST and WCPSTC matrices may have influenced the extraction and quantification method. Possible variations in dry matter, fat and protein contents of whey cheeses used in this study might also have had some influence on the results, even though they belonged to the same batch.

### 3.4. Lactic Acid and Lactose Contents during Storage

Variation in lactose and lactic acid concentration of whey cheese formulations during storage is presented in Figure 2. Non-inoculated whey cheese concentrations of lactose and lactic acid were (mg g^−1^) 33.86 ± 1.96 and 2.05 ± 0.02, respectively.

Consumption of lactose and concomitant production of lactic acid occurred in all probiotic whey cheese matrices, with values of lactose decreasing on average from 31 mg g^−1^ to 25 mg g^−1^ at the end of the storage period; the results are in agreement with those found by other authors [21]. No significant differences were found between matrices regarding lactose consumption. On the other hand, from an initial mean value of 5.0 mg g^−1^, lactic acid increased to ca. 20 mg g^−1^ in most formulations, which represents an increase in a lactic acid concentration similar to that found by Madureira et al. [7] and Pescuma et al. [40]. Lactic acid evolution during storage exhibits some differentiation among the group of matrices, revealing that matrix WCPSTC’s production of lactic acid lowered significantly in comparison to the others from day 14 onward due to sodium citrate’s influence in this whey cheese.

A good correlation was found between lactose and lactic acid concentrations in all whey cheese matrices, ranging from −0.840 (*p* < 0.01) for the WCP matrix and −0.933 (*p* < 0.01) for WCPST whey cheese.

### 3.5. Textural Evaluation of Whey Cheeses

Textural parameters on the day of manufacture and on the last day of storage (21 d) are depicted in Figure 3, where results of non-inoculated whey cheese and the four formulations tested are presented. Considering that the products in the study (spreadable cheese) are defined as semisolids, textural attributes such as chewiness, springiness and fracturability were not calculated.

Textural results are intimately associated with manufacturing conditions and, above all, the chemical composition of the final product. Correlations were already established between pH, dry matter, protein and fat contents and textural properties of cheeses [41]. Considering this, overall, we observed three distinct textural profiles: one pertaining to traditional and non-inoculated whey cheese (WC); the second group consisting of WCP and WCPS whey cheeses; and finally, a profile of WCPST and WCPSTC matrices. A little variation in textural properties was observed between the day of manufacture and the final day of storage for most matrices.

Whey cheese WC (control) was not subjected to any further processing, which explains most of the significant differences found between it and the other matrices. The other whey cheeses were pasteurized and mixed thoroughly, which in turn resulted in creamy and spreadable products. However, some variations in texture within the group are observable, probably due to their intrinsic chemical composition.

Hardness, a measure of force needed to attain a given deformation, and gumminess, an indicator of force needed to disintegrate the food to a state ready for swallowing, have been known to express a positive correlation [42]. In fact, a very similar tendency of both texture attributes was observed between whey cheeses with a high degree of correlation (r = 0.981; *p* < 0.01). The control sample (WC), not suffering any processing or additive incorporation, presented a significantly higher value of hardness (Figure 3a) and gumminess (Figure 3c) in comparison to the others. On the other hand, WCPST and WCPSTC presented the lowest values of the group in both texture attributes. WCP and WCPS showed intermediate values, significantly different from the other matrices, which can be attributed to the lower fat content of these samples in comparison to WCPST and WCPSTC. As suggested by Zheng et al. [43], a significant negative correlation between total fat content and hardness was also observed in this study (r = −0.754; *p* < 0.01).

Cohesiveness reflects an inverted scenario with firmer whey cheese revealing a less cohesive matrix. It is known that traditional Portuguese whey cheese “Requeijão” has a lumpy and friable texture [44], and as such, it was expected that significant differences would be observable between control and processed whey cheese samples (Figure 3d).

Adhesiveness profile (Figure 3b) revealed a good correlation with both fat content (r = 0.780; *p* < 0.01) and hardness (r = −0.841; *p* < 0.01) in agreement with the findings of Brighenti et al. [45] referring again to variations in chemical composition as the main reason for observed differences.

In conclusion, the textural properties of the optimized matrix (WCPSTC), in direct comparison to the other matrices (including control), revealed a significant decrease in hardness, adhesiveness and gumminess, which translated into a smoother and creamier product while, simultaneously, not affecting its cohesiveness, which in turn relates to the good spreadability of the optimized product.

## 4. Final Remarks

The main objectives set out to be achieved within this research framework were successfully completed. Firstly, SCWP inclusion in whey cheese matrix at 5% (*w*/*w*) did not affect *B. animalis* BB12 and *L. rhamnosus* R-11 viable cell numbers evolution, and at chemical and texture levels, only minor changes were detected. SCWP incorporation was responsible for a slight increase in protein and dry matter contents, which was reflected mainly in positive changes in texture profile (reduced hardness and gumminess). Secondly, sensorial improvement of probiotic whey cheese through the addition of thyme EO and sodium citrate was revealed to be effective in adding organoleptic value and acidity control, respectively. Lastly, probiotic co-culture comprised of *Bifidobacterium animalis* subsp. *lactis* BB-12^®^ and *Lactobacillus rhamnosus* Rosell^®^-11 managed to maintain a high concentration of viable cell numbers throughout storage (above 10^7^ CFU g^−1^), regardless of additives or SCWP incorporation. This research effort provided effective and innovative strategies to overcome the lack of organoleptic quality associated with probiotic whey cheese while successfully incorporating in its matrix SCWP, an environmentally impactful dairy by-product contributing to waste minimization and sustainable reuse of an important nutrient-effective by-product.

## Figures and Tables

**Figure 1 foods-11-02698-f001:**
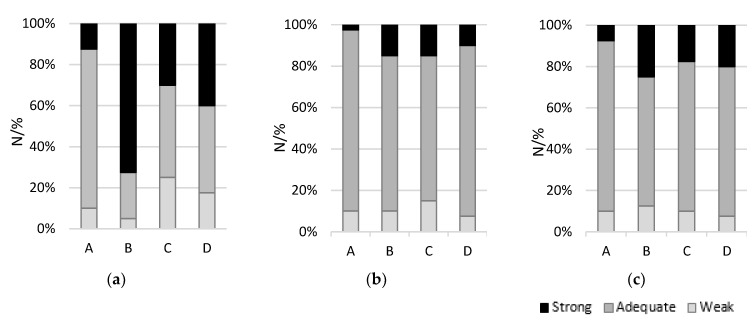
Intensity scores of sensory attributes: (**a**) Aroma; (**b**) Saltiness; (**c**) Acidity. Formulations: A (thyme); B (thyme + garlic + sage); C (thyme + sage); D (thyme + garlic). Base formulation composed of whey cheese, 85% (*w*/*w*); SCWP, 5% (*w*/*w*) and probiotic culture, 10% (*v*/*w*). Essential oils were added at 0.025% (*w*/*w*) concentration.

**Figure 2 foods-11-02698-f002:**
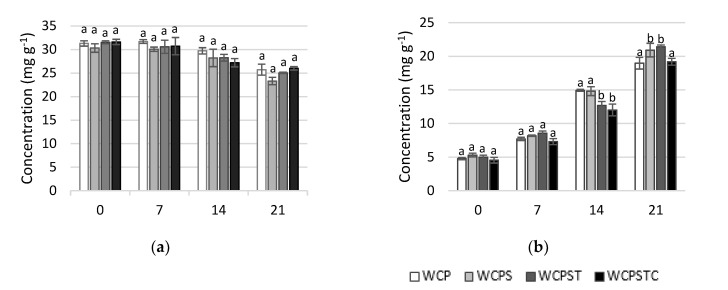
Lactose (**a**) and lactic acid (**b**) concentration of whey cheeses matrices during storage. WCP (control); WCPS (with SCWP); WCPST (with SCWP and thyme EO); WCPSTC (with SCWP, thyme EO and sodium citrate). ^a,b^ Means within the same day of storage labeled with different superscripts differ significantly (*p* < 0.05).

**Figure 3 foods-11-02698-f003:**
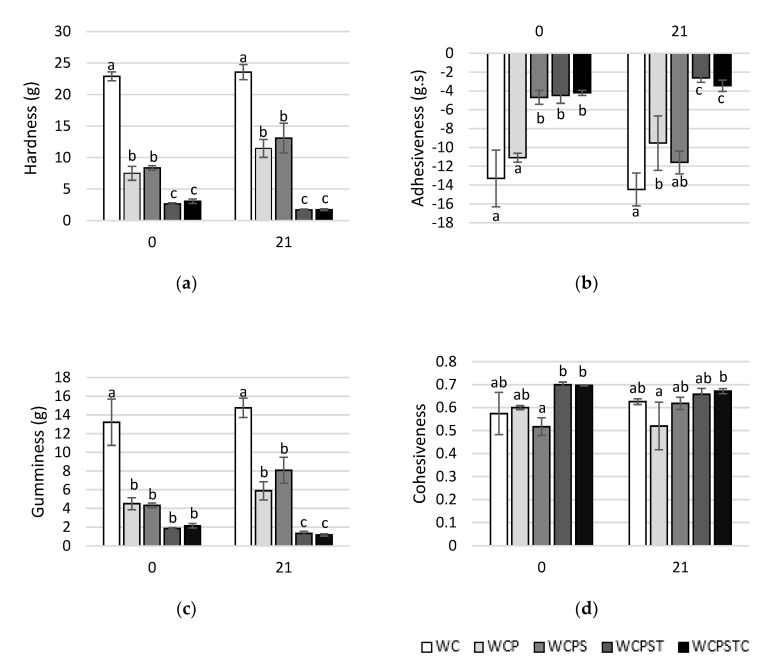
Textural attributes of whey cheese matrices at beginning and end of storage. Parameters: hardness (**a**); adhesiveness (**b**); gumminess (**c**); cohesiveness (**d**). WC (non-inoculated); WCP (control); WCPS (with SCWP); WCPST (with SCWP and thyme EO); WCPSTC (with SCWP, thyme EO and sodium citrate). ^a,b,c^ Means within the same day of storage labeled with different superscripts differ significantly (*p* < 0.05).

**Table 1 foods-11-02698-t001:** Minimum inhibitory concentrations (MIC) of essential oils and sodium citrate against food pathogens, contaminant bacteria and probiotic bacteria. Results are expressed in concentration of mg mL^−1^.

Species of Bacteria	Essential Oil/Sodium Citrate				
	Sage	Garlic	Thyme	Sodium Citrate	Thyme + Sodium Citrate
*B. cereus*	>1	>1	0.5	ND	0.5 + 16.0
*L. monocytogenes*	>1	0.075	0.25	ND	0.25 + 8.0
*S. aureus*	>1	>1	0.125	ND	0.25 + 8.0
*E. coli*	>1	>1	0.25	ND	0.25 + 8.0
*S. enteritidis*	>1	>1	0.5	ND	0.25 + 8.0
*B. animalis* BB-12	>1	>1	0.5	16.0	0.5 + 16.0
*L. rhamnosus* R-11	>1	>1	0.5	32.0	1.0 + 32.0

ND—not determined.

**Table 2 foods-11-02698-t002:** Aromatized whey cheese sensory results (average ± standard deviation).

Sensory Traits	Aromatized Formulations of Whey Cheese			
	A	B	C	D
Flavor	6.95 ± 1.34 ^a^	5.45 ± 2.17 ^b^	5.53 ± 1.84 ^b^	5.78 ± 2.02 ^b^
Texture	7.13 ± 1.38 ^a^	6.85 ± 1.49 ^a^	6.75 ± 1.20 ^a^	6.88 ± 1.44 ^a^
Overall acceptance	7.00 ± 1.18 ^a^	5.65 ± 1.96 ^b^	6.00 ± 1.48 ^b^	5.98 ± 1.92 ^b^

Formulations: A (thyme); B (thyme + garlic + sage); C (thyme + sage); D (thyme + garlic). Base formulation composed of whey cheese, 85% (*w*/*w*); SCWP, 5% (*w*/*w*) and probiotic culture, 10% (*v*/*w*). Essential oils were added at 0.025% (*w*/*w*) concentration. Results presented in hedonic scale base 9, where 1 = “very bad”, 9 = “very good”. ^a,b^ Means within the same line labeled with different superscripts differ significantly (*p* < 0.05).

**Table 3 foods-11-02698-t003:** Evolution of viable counts (log CFU g^−1^) and physicochemical parameters throughout storage period (average ± standard deviation).

Strain	Storage Time (Day)	Whey Cheese Matrix			
		WCP	WCPS	WCPST	WCPSTC
*Bifidobacterium animalis* subsp. *lactis* BB-12	0	8.19 ± 0.10 ^a^	8.30 ± 0.18 ^a^	7.24 ± 0.07 ^b^	7.38 ± 0.27 ^b^
	7	8.40 ± 0.17 ^a^	8.39 ± 0.13 ^a^	7.24 ± 0.07 ^b^	7.11 ± 0.26 ^b^
	14	8.78 ± 0.08 ^a^	8.99 ± 0.44 ^a^	7.36 ± 0.05 ^b^	7.31 ± 0.05 ^b^
	21	8.93 ± 0.07 ^a^	8.69 ± 0.19 ^a^	7.64 ± 0.11 ^b^	7.45 ± 0.05 ^b^
*Lactobacillus rhamnosus* Rosell-11	0	7.99 ± 0.05 ^a^	8.24 ± 0.19 ^a^	7.25 ± 0.09 ^b^	7.20 ± 0.10 ^b^
	7	8.46 ± 0.26 ^a^	8.36 ± 0.09 ^a^	7.31 ± 0.09 ^b^	7.26 ± 0.10 ^b^
	14	8.64 ± 0.18 ^a^	8.28 ± 0.17 ^a^	7.34 ± 0.07 ^b^	7.35 ± 0.13 ^b^
	21	8.71 ± 0.15 ^a^	8.46 ± 0.14 ^ab^	7.98 ± 0.11^c^	8.06 ± 0.10 ^bc^
pH	0	5.72 ± 0.01 ^a^	5.66 ± 0.02^a^	6.02 ± 0.01 ^b^	6.25 ± 0.00 ^c^
	7	5.59 ± 0.01 ^a^	5.49 ± 0.04 ^a^	5.87 ± 0.01 ^b^	6.21 ± 0.02 ^c^
	14	4.80 ± 0.02 ^a^	5.06 ± 0.08 ^b^	5.39 ± 0.01 ^c^	5.57 ± 0.07 ^d^
	21	4.79 ± 0.00 ^a^	4.87 ± 0.08 ^a^	4.77 ± 0.01 ^a^	5.05 ± 0.01 ^b^
Titratable acidity (% Lactic acid)	0	0.42 ± 0.00 ^a^	0.52 ± 0.03 ^a^	0.49 ± 0.01 ^a^	0.43 ± 0.01 ^a^
	7	0.55 ± 0.02 ^a^	0.65 ± 0.04 ^a^	0.64 ± 0.02 ^a^	0.53 ± 0.03 ^a^
	14	0.93 ± 0.02 ^a^	0.84 ± 0.07 ^ab^	0.83 ± 0.03 ^ab^	0.79 ± 0.03 ^b^
	21	1.10 ± 0.07 ^ab^	1.17 ± 0.10 ^ab^	1.24 ± 0.01 ^b^	1.07 ± 0.01 ^a^

WCP (control); WCPS (with SCWP); WCPST (with SCWP and thyme EO); WCPSTC (with SCWP, thyme EO and sodium citrate). ^a,b,c,d^ Means within the same line labeled with different superscripts differ significantly (*p* < 0.05).

**Table 4 foods-11-02698-t004:** Physicochemical composition of whey cheeses matrices (average ± standard deviation).

Parameter	Whey Cheese Matrix				
	WC	WCP	WCPS	WCPST	WCPSTC
Dry matter (%)	27.02 ± 0.11 ^a^	25.17 ± 0.04 ^b^	27.35 ± 0.04 ^a^	25.92 ± 0.20 ^c^	25.64 ± 0.13 ^bc^
Ash (%)	3.32 ± 0.07 ^a^	3.33 ± 0.01 ^a^	3.08 ± 0.08 ^ab^	2.48 ± 0.01 ^c^	2.81 ± 0.02 ^b^
Fat (%)	6.13 ± 0.31 ^a^	5.71 ± 0.10 ^b^	6.45 ± 0.31 ^a^	8.03 ± 0.09 ^c^	7.59 ± 0.03 ^c^
Protein (%)	11.10 ± 1.06 ^ab^	10.84 ± 0.14 ^a^	12.66 ± 0.08 ^b^	11.26 ± 0.25 ^ab^	10.78 ± 0.08 ^a^

WC (non-inoculated); WCP (control); WCPS (with SCWP); WCPST (with SCWP and thyme EO); WCPSTC (with SCWP, thyme EO and sodium citrate). ^a,b,c^ Means within the same line labeled with different superscripts differ significantly (*p* < 0.05).

## Data Availability

Data is contained within the article.

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
