# Peer review of "Development and Characterization of a Novel Sustainable Probiotic Goat Whey Cheese Containing Second Cheese Whey Powder and Stabilized with Thyme Essential Oil and Sodium Citrate"

_foods, 2022, doi:10.3390/foods11172698_

Round 1

Reviewer 1 Report

From what period was goat's milk used?,

What was the physicochemical composition?

Due to the diversity of composition, it is difficult to refer to the whole study.  It cannot be representative. Little data on composition 

In what amount probiotics were added per gram of product, unclear

Author Response

Response to reviewer 1 comments

Point 1: From what period was goat's milk used?

Response 1: The authors acknowledge that the period of goat milk used for whey cheese production would be useful information. Unfortunately, the authors do not have that information, they can only add that the whey cheese used in the study was manufactured between September and October of 2019.

Point 2: What was the physicochemical composition?

Response 2: In fact, the physicochemical composition would add more complexity and robustness to the study. Nevertheless, the authors consider that the focus of the study being in whey cheese and second whey cheese, the characterization of the raw material, goat milk, was not included in the study design.

Point 3: Due to the diversity of composition, it is difficult to refer to the whole study.  It cannot be representative. Little data on composition

Response 3: Complementing the response given in the previous question/comments, the authors would like to add that although they do not possess any information regarding the goat milk used to produce whey cheese, second whey cheese chemical composition was added to the manuscript to complement the study, lines 131-133.

Point 4: In what amount probiotics were added per gram of product, unclear

Response 4: The authors are thankful for the reviewer’s suggestions and this issue was addressed in the section of Materials and Methods 2.3., lines 142-143, adding more information concerning the volume and concentration of probiotic cultures.

Reviewer 2 Report

The manuscript entitled "Development and characterization of a novel sustainable probiotic goat whey cheese incorporating second cheese whey powder and stabilized with thyme essential oil and citric acid" was reviewed. In this study the authors have attempted to introduce a strategy for using an industrial by-product (second cheese whey) as an ingredient in the manufacture of a new whey cheese with improved organoleptic and physico-chemical properties. Although the overall issue is interesting, and the authors have done different analytical and microbiological tests, the study does not have considerable novelty. The authors have purchased a commercial whey cheese, and blended it with second cheese whey powder and thyme essential oil, sodium citrate, and probiotic cultures. Then, some microbiological, and physiochemical tests have been carried out on the final product. In addition, there is no logical reason for performing some assays. For example, the authors have evaluated the antimicrobial activity of essential oils via microdilution method in the culture media that could not be generalized in the product. It is expected, when the authors concentrate on safety and antimicrobial activity of different ingredients, they evaluate the effects in the final product. Therefore, the authors are recommended to add antimicrobial data from in situ assays.  

The manuscript needs some technical and English corrections as below:            

Title:

 Development and characterization of a novel sustainable probiotic goat whey cheese incorporating second cheese whey powder and stabilized with thyme essential oil and citric acid: It is recommended that "incorporating" is replaced with "containing". Also, please write "sodium citrate" instead of "citric acid".

Abstract:                                                      

L16- All abbreviates should be defined for the first time. For example: WCPSTC

L16: The fat content is usually expressed as dry basis.

L18-L19: Please rewrite the sentences.

L-21: Safety? What safety evaluations have been done? Please mention them in Abstract.

L-49: Please change "Research has" to "Researches have"

Materials and Methods:

L93-97: Did you use L-Cys for Lactobacillus culture? How did you prepare anaerobic conditions?

L-110-L113: Please rewrite the concentration process for SCW preparation.

2.2: Second cheese whey processing: What were the characteristics of second cheese whey used in the study? Did the authors have any information on spore contamination of the used second cheese whey?

L-118: 2.3. Probiotic whey cheese process: I think it could be better if it is replaced with: "Manufacture of probiotic whey cheese"  

L-121-123: How did you pasteurize the mixture of Requeijao cheese and SCWP?

L-125-L126: For probiotic inoculation, the initial concentration of bacterial cultures is very important, and it should be noted as bot CFU/g and volumetric ratio. A query on cheese formulation is about the proportion of bacterial cultures. Is it logical to mention that bacterial cultures comprised 10% of final product?     

 2.4. Microbiological analysis: Did the authors first calculate MIC for essential oil and sodium citrate before determining appropriate concentration to be added to the products (0.025%, and 0.8 %)?

Results and Discussion

3.1: There is no novelty in the evaluation of antimicrobial activity of essential oils against contaminant bacteria. What was the necessity for determining antimicrobial activity of EOs against contaminant bacteria? It could be very interesting if the authors had contaminated the product and performed in situ antimicrobial evaluation. Please add the in situ data, if you have.   

L-263-270: Please rewrite the sentences. It is recommended to use the PAST tense of the verbs.

Table 2. What is the basis for scores? From 10? It should be mentioned.

Page 12, Figure 2. What do you mean EVOLUTION? Better word usage is recommended.

3.5. Textural evaluation of whey cheeses: More discussion is needed for the clarification of possible impact of the treatments applied by the authors on textural properties. Comparison to the previous studies is useful, but not enough.  

Author Response

Response to reviewer 2 comments

General comment: The manuscript entitled "Development and characterization of a novel sustainable probiotic goat whey cheese incorporating second cheese whey powder and stabilized with thyme essential oil and citric acid" was reviewed. In this study the authors have attempted to introduce a strategy for using an industrial by-product (second cheese whey) as an ingredient in the manufacture of a new whey cheese with improved organoleptic and physico-chemical properties. Although the overall issue is interesting, and the authors have done different analytical and microbiological tests, the study does not have considerable novelty. The authors have purchased a commercial whey cheese, and blended it with second cheese whey powder and thyme essential oil, sodium citrate, and probiotic cultures. Then, some microbiological, and physiochemical tests have been carried out on the final product. In addition, there is no logical reason for performing some assays. For example, the authors have evaluated the antimicrobial activity of essential oils via microdilution method in the culture media that could not be generalized in the product. It is expected, when the authors concentrate on safety and antimicrobial activity of different ingredients, they evaluate the effects in the final product. Therefore, the authors are recommended to add antimicrobial data from in situ assays. 

Response: The authors would like to thank the reviewer for the general comments. A deep revision was done throughout the manuscript according to the Reviewers comments in order to improve the clarity and quality of the manuscript. The issue of the antimicrobial activity was addressed and explained below, response 14.

Point 1: Development and characterization of a novel sustainable probiotic goat whey cheese incorporating second cheese whey powder and stabilized with thyme essential oil and citric acid: It is recommended that "incorporating" is replaced with "containing". Also, please write "sodium citrate" instead of "citric acid".

Response 1: According to the reviewer’s recommendation, the authors have changed the title.

Point 2: L16- All abbreviates should be defined for the first time. For example: WCPSTC

Response 2: According to the reviewer’s suggestion, the authors have altered the abstract and the rest of the manuscript in order to define the abbreviation of the matrices from the first mention.

Point 3: L16: The fat content is usually expressed as dry basis.

Response 3: The authors acknowledge that as the reviewer mentioned, the fat content is frequently expressed as dry basis. However, the authors due to time constraints for the revision of this manuscript did not alter fat content value expression, which is still in % (w/w) throughout the manuscript.

Point 4: L18-L19: Please rewrite the sentences.

Response 4: According to the reviewer’s suggestion these sentences of the abstract were rewritten.

Point 5: L-21: Safety? What safety evaluations have been done? Please mention them in Abstract.

Response 5: The authors agree that no significant safety evaluations were done, only antimicrobial activity of essential oil added, but then the impact in the final product was not investigated. As such, it was removed from abstract the mention of enhanced product safety.

Point 6: L-49: Please change "Research has" to "Researches have"

Response 6: According to the reviewer’s suggestion this line was changed.

Point 7: L93-97: Did you use L-Cys for Lactobacillus culture? How did you prepare anaerobic conditions?

Response 7: Yes, the authors used L-cysteine to enhance growth of probiotic bacteria, both Bifidobacterium animalis subsp. lactis BB-12® (BB-12) and Lactobacillus rhamnosus Rosell®-11 (R-11) and a Don Whitley anaerobic chamber was used to achieved anaerobic conditions, lines 104-107 of the reviewed manuscript.

Point 8: L-110-L113: Please rewrite the concentration process for SCW preparation.

Response 8: According to the reviewer’s recommendation this section of the materials and methods was rewritten.

Point 9: 2.2: Second cheese whey processing: What were the characteristics of second cheese whey used in the study? Did the authors have any information on spore contamination of the used second cheese whey?

Response 9: The authors acknowledge that pertinent information was missing from this section and thank the reviewer for pointing that out. As such, lines 129-132 add information regarding sterilization of second cheese whey powder, that in turn resulted in a microbiologically safe product (no contamination was detected in the study). Also, physiochemical composition of the second cheese whey powder was added to complement the study.

Point 10: L-118: 2.3. Probiotic whey cheese process: I think it could be better if it is replaced with: "Manufacture of probiotic whey cheese"  

Response 10: According to the reviewer’s recommendation this section title was modified.

Point 11: L-121-123: How did you pasteurize the mixture of Requeijao cheese and SCWP?

Response 11: According to lines 135-137, Requeijão and SCWP at 5% (w/w) were pasteurized for 30 min at 100 °C using a Bimby® TM6 (Vorwerk Thermonix, Germany).

Point 12: L-125-L126: For probiotic inoculation, the initial concentration of bacterial cultures is very important, and it should be noted as bot CFU/g and volumetric ratio. A query on cheese formulation is about the proportion of bacterial cultures. Is it logical to mention that bacterial cultures comprised 10% of final product?   

Response 12: The authors agree with the reviewer's comment regarding probiotic inoculation. In that regard, lines 140-143 were added in an effort to clarify the inoculation process, including initial concentration of the probiotic cultures, BB-12 and R-11.

Point 13: 2.4. Microbiological analysis: Did the authors first calculate MIC for essential oil and sodium citrate before determining appropriate concentration to be added to the products (0.025%, and 0.8 %)?

Response 13: Yes, lines 180-190 describe the MIC method applied to find optimum concentration of essential and sodium citrate. In the result section lines 271-303 present the results that the author found and led them to conclude that the optimum concentration of EO and sodium citrate was indeed 0.025% and 0.8% respectively. Furthermore sensory analysis was conducted to confirm that concentration of EO at 0.025% was not overpowering or too weak from an organoleptic standpoint.

Point 14: 3.1: There is no novelty in the evaluation of antimicrobial activity of essential oils against contaminant bacteria. What was the necessity for determining antimicrobial activity of EOs against contaminant bacteria? It could be very interesting if the authors had contaminated the product and performed in situ antimicrobial evaluation. Please add the in situ data, if you have.  

Response 14: The authors understand and agree with the reviewer comment on antimicrobial activity of essential oils. Despite being interesting the inoculation of the product with contaminant bacteria to confirm antimicrobial activity of thyme EO, this study focused on the technological and sensorial aspects of development of a probiotic whey cheese, so unfortunately no in situ data are available. The MIC tests of EO against contaminant bacteria were conducted mainly to confirm antimicrobial activity of EO, which in turn would putatively increase microbial safety of the final product, even though this increased safety was not evaluated. In summary the authors felt that would add depth to the study and as such, the addition of thyme EO would not only be seen as tool to enhance organoleptic features (aroma and taste), although it was the main purpose, but also as functional addition for microbial safety.

Point 15: L-263-270: Please rewrite the sentences. It is recommended to use the PAST tense of the verbs.

Response 15: According to the reviewer’s recommendation these lines were rewritten.

Point 16: Table 2. What is the basis for scores? From 10? It should be mentioned.

Response 16: The basis of the score was hedonic scale from 1 to 9 as stated in lines 253-254 of sensory analysis description in the section of materials and methods. However, the authors agree with the reviewer that this information should be present closer to the results, as such, according to reviewer comment the mention of score scale was added to the caption of Table 2.

Point 17: Page 12, Figure 2. What do you mean EVOLUTION? Better word usage is recommended.

Response 17: The authors acknowledge some technical and English language shortcomings in the text and as such have made a complete overhaul of the manuscript in terms of word usage hoping to achieve a more concise and adequately written study.

Point 18: 3.5. Textural evaluation of whey cheeses: More discussion is needed for the clarification of possible impact of the treatments applied by the authors on textural properties. Comparison to the previous studies is useful, but not enough. 

Response 18: According to the reviewer comment, the author have added a new paragraph to this section, lines 462-466, that hopefully will adequately complement the discussion of textural properties of the probiotic whey cheese matrices.

Reviewer 3 Report

Dear Authors,

the submitted paper is very original and interesting from the perspective of dairy industry by-product recovery and the circular economy topic.

Some changes and improvements need to be made to the paper, in particular, I would ask you to check for more up-to-date bibliographic references than the ones you have included, and in any case to enrich the introduction. I note an excessive number of references to previous papers by one or more authors in this paper.

Also, check the captions of figures and tables , they should be consistent with the journal guidelines, and check that everything fits on the page; often figures and tables are out of the margins.

Otherwise, I suggest the following:

Line 10: please, with reference to the paper guidelines, add all the requested details for the reference person

Line 278: Table 1 is not in the page and it should be formatted with regard to the paper guideline

Line 300: Figure 1 has a poor resolution. Please, provide a better quality image

Line 318: Please, check the caption, it does not seem to be coherent with the paper guideline. Delete (-)

Line 327: Please, authors explain better how much the founded values are different (signficantly) from those found in the cited references

Line 366: Please, refer to previous comments on table captions

Line 385: Please, check figure 2, with reference to their alignation in the text, and the rules for the captions

Line 407: see Line 385 (figure 3)

Line 465: please, check the numbered list for references, it seems to be present a double number for each reference

Author Response

Response to reviewer 3 comments

General comment: The submitted paper is very original and interesting from the perspective of dairy industry by-product recovery and the circular economy topic. Some changes and improvements need to be made to the paper, in particular, I would ask you to check for more up-to-date bibliographic references than the ones you have included, and in any case to enrich the introduction. I note an excessive number of references to previous papers by one or more authors in this paper. Also, check the captions of figures and tables , they should be consistent with the journal guidelines, and check that everything fits on the page; often figures and tables are out of the margins.

Response: The authors would like to thank the reviewer for the general comments. A deep revision was done throughout the manuscript according to the Reviewers comments in order to improve the clarity and quality of the manuscript. Although the authors agree with the reviewer comment on the bibliographic references, most of them are recent and despite having made an effort to find more up-to-date references relevant to this study, unfortunately none were found. Regarding the issue of the authors previous studies being mentioned throughout this study, the authors feel that considering that the present study follows and adds relevant information to the same line of investigation, the references are adequate. All formatting issues were addressed and are now consistent with the journal guidelines.

Point 1: Line 10: please, with reference to the paper guidelines, add all the requested details for the reference person

Response 1: According to reviewer recommendation, relevant information of the reference person was added.

Point 2: Line 278: Table 1 is not in the page and it should be formatted with regard to the paper guideline

Response 2: According to reviewer suggestion, the table was reformatted to fit the paper guidelines and placed correctly in the manuscript.

Point 3: Line 300: Figure 1 has a poor resolution. Please, provide a better quality image

Response 3: The authors have corrected this issue, providing a better quality image of Figure 1.

Point 4: Line 318: Please, check the caption, it does not seem to be coherent with the paper guideline. Delete (-)

Response 4: The authors are thankful for the reviewer’s suggestions and all caption, figures and tables throughout the manuscript are now consistent with paper guidelines.

Point 5: Line 327: Please, authors explain better how much the founded values are different (signficantly) from those found in the cited references

Response 5: According to reviewer suggestion the newly added lines 345-349 now explain in greater detail the difference between viable cell concentration results in the present study and the cited reference.

Point 6: Line 366: Please, refer to previous comments on table captions

Response 6: The authors are thankful for the reviewer’s suggestions and all caption, figures and tables throughout the manuscript are now consistent with paper guidelines.

Point 7: Line 385: Please, check figure 2, with reference to their alignation in the text, and the rules for the captions

Response 7: The authors are thankful for the reviewer’s suggestions. All caption, figures and tables throughout the manuscript have been formatted to fit the paper guidelines and alignment with the text.

Point 8: Line 407: see Line 385 (figure 3)

Response 8: The authors are thankful for the reviewer’s suggestions and all caption, figures and tables throughout the manuscript are now consistent with paper guidelines.

Point 9: Line 465: please, check the numbered list for references, it seems to be present a double number for each reference

Response 9: Indeed, there was duplication of the number of each reference in the reference list and reference 45 was mistakenly duplicated in the text, not mentioning reference 46. The authors thank the reviewer for pointing this error and in the revised manuscript this has been corrected. 

Round 2

Reviewer 2 Report

The authors have responded to the queries and made the corrections.